# We're Here to Help: Company Image Repair and User Perception of Data Breaches

Zahra Hassanzadeh*
Carleton University
Ottawa, Canada

Sky Marsen†
Flinders University
Adelaide, Australia

Robert Biddle‡
Carleton University
Ottawa, Canada

## ABSTRACT

Data breaches involve information being accessed by unauthorized parties. Our research concerns user perception of data breaches, especially issues relating to accountability. A preliminary study indicated many people had weak understanding of the issues, and felt they themselves were somehow responsible. We speculated that this impression might stem from organizational communication strategies. We therefore compared texts from organizations with those from external sources, such as the news media. This suggested that organizations use well-known crisis communication methods to reduce their reputational damage, and that these strategies align with repositioning of the narrative elements involved in the story. We then conducted a quantitative study, asking participants to rate either organizational texts or news texts about breaches. The findings of this study were in line with our document analysis, and suggest that organizational communication affects the users' perception of victimization, attitudes in data protection, and accountability. Our study suggests some software design and legal implications to support users protecting themselves and developing better mental models of security breaches.

**Index Terms:** Security and privacy—Social aspects of security and privacy; Human-centered computing—User models

## 1 INTRODUCTION

A data breach is a successful malicious attack which leads to the compromise or the loss of data [18]. Personally Identifiable Information (PII) is often stored in organization databases, and if disclosed is at risk of misuse. Depending on the size, scale, and type of stolen information, the potential consequences of a data breach can be huge. A data breach can put people at risk of identity theft, which often happens through fraudulent use of existing accounts like credit cards, online accounts, and insurance. It can also lead to financial loss and emotional distress [22].

Despite the increased awareness of organizations and great emphasis by experts on security mechanisms, many organizations still maintain insufficient security practices for data collection, processing, and storage, so are unable to prevent data breaches and consequent misuse of the data. Several recent occurrences follow this pattern, and data breaches at major companies, like Equifax, have exposed a massive number of consumers' records [17].

Although such events have become commonplace, there appears to be little indication that end-users feel urgency about holding companies to account. A 2016 study reports that by far most consumers kept doing business with companies after breaches [1], and some high-profile commentary suggests "breach fatigue" has "set a new normal and instill a sense of fatalism — and complacency" [16]. In a small preliminary study, we even found that participants often

---

*e-mail: zahra.hassanzadeh@carleton.ca
†e-mail: sky.marsen@flinders.edu.au
‡e-mail: Robert.Biddle@carleton.ca

thought that they themselves were somehow responsible for data breaches.

According to Coombs [5], the reputation of a company is based on the evaluation customers make about it. Customer evaluations can be affected by the behavior of a company when a crisis like a data breach happens. So, due to the significant financial loss and reputational damage caused by data breaches, companies try to reduce the damage using communication strategies in the after-breach notifications [9, 19]. The crisis response strategies aim to reduce the negative effects of the crisis by changing the level of crisis responsibility. For example, if a company frames themselves as victims of the situation and therefore positioned in what crisis communication theorists call the "victim cluster", they are likely to incur little blame for the crisis [5].

User understanding of data breach incidents is important because it allows development of mental models to support reasoning about behavior and accountability [3]. The goal of our research is to explore how breached companies and the news media communicate with users, and how that might affect users' perception of a data breach incident. To do so, we apply *Image Repair Theory (IRT)* [2] and a *narrative-semiotics* method [8] to the analysis of Equifax crisis communications to see how this incident is reported in the company press releases and the news. We first conducted a communication study based on collected data from 58 stories related to this security breach crisis. We then conducted a questionnaire study with 100 participants testing the influence of companies' notifications and news on the general public. To the best of our knowledge, testing communication strategies' influence on users mental models of a data breach is original, and it shows HCI efforts on building user understanding of security can be undermined by organizational communication. Our results also suggest a need for the improvement of software design, and dedicated attention of communication professionals and legal scholars to the notifications created during and after a data breach.

## 2 BACKGROUND

Data breaches are a severe threat to both organizations and consumers, and are increasingly common. When a data breach happens, the organization facing this crisis is required to inform the legal authorities, and it also needs to notify all the affected and potentially affected consumers.

For these reasons, data breaches are now understood as a *crisis* for an organization, and there is an established body of knowledge about how organizations should communicate about crises. There are several theories on effective communication strategies during crises. In our work we used *Image Repair Theory (IRT)*, and also a narrative-semiotic method.

### 2.1 Image repair

IRT, introduced by William Benoit [2] is a well-established framework, and it can be used by practitioners to design messages during a crisis. It can also be used by critics to evaluate the created messages. The key concept in this theory is to understand the nature of a complaint or attack. An attack has two components: an offensive act, and the accusation of responsibility for an action. According

to Benoit, the image repair strategies can be categorized into five broad categories: denial, evasion of responsibility, reducing the offensiveness of event, corrective action, and mortification. Denial is a general approach to image repair, and it is about rejecting responsibility. Evasion of responsibility includes four sub-strategies: provocation (response to someone else's action), defeasibility (lack of information or control over the situation), accident (did not mean it to happen), and good intention. Reducing the offensiveness of an event also involves a list of potential response strategies: bolstering (reminds of good traits), minimization (claim that act was not serious), differentiation (reduce offensiveness of the act), transcendence (place the act in more favorable context), attack accuser (challenge the credibility of accuser), and compensation (reimburse for the act). Corrective action is about restoring a situation or promise that the act will not happen again, and mortification is asking for forgiveness.

## 2.2 Narrative-semiotic approach

Semiotics involves the study of signs, and semioticians believe that communication is symbolic and ambiguous, and it happens through perceptual or linguistic signs between interlocutors. The narrative-semiotic method finds common patterns in stories [14]. The strategies for examining storytelling help to make sense of what the narrator has perceived and experienced. This can reveal conflicts and changes during a crisis [11].

Based on news framing theory, media and organizations use different features in their messages to frame the crisis [21]. We can use narrative-semiotic to see how these framings are different. This method is valuable for understanding a data breach situation, and the goals and motivations of different narrators of a data breach story. It can help researchers to study a failure of an organization and assess whether the decisions made by the organizations were appropriate or not [15].

We can divide the narrative-semiotic approach into two aspects: the *narrative trajectory* which is the sequence of events and actions that create a story, and the *narrative schema* [15].

The narrative schema consists of the six categories of agents known as the actantial model. The actantial model, developed by A.J. Greimas [8], can be used to break an action down into six positions or *actants*:

1. The *sender* includes agents who direct the action of subject towards an object.

2. The *subject* consists of the leading performers aiming at a desired goal or object.

3. The *object* includes the desired goal and objectives.

4. The *receiver* category consists of the agents who benefit if a desired goal is achieved.

5. The *helper* includes agents who assist in achieving the desired goal, like experts who aid the subject.

6. The *opponent* includes agents who hinder the achievement of the desired goal, for example, adversaries, lack of knowledge or ability, and ineffective tools.

Narrative-semiotic can be used to identify the agents, their actions, and their discourses. Different actions reflect different points of view, which is why it is vital to compare different texts on the same subject. A series of events can be told differently when the narrator changes. Narrators position the agents through the way they tell the story. So the narrator has a pivotal role in interpreting the actions and highlighting the patterns in their interaction [13]. Comparing the narrative structure of different texts can reveal how companies narrate a crisis like a data breach, and how their narrative can be different from the narratives produced by media organizations.

The way a story is told using strategies like IRT might change readers' mental model of a data breach, in other words, the way they

Table 1: Equifax: number of stories per publication and timeline (PR= Press release, GAO= Government Accountability Office, R= The Register, NYT= New York Times, WP= The Washington Post, G= The Guardian, T= Total)

| Timeline | PR | GAO | R | NYT | WP | G | T |
|---|---|---|---|---|---|---|---|
| September, 2017 | 2 | 0 | 9 | 6 | 1 | 6 | 24 |
| October, 2017 | 2 | 0 | 7 | 0 | 0 | 1 | 10 |
| November, 2017 | 0 | 0 | 2 | 0 | 0 | 0 | 2 |
| January, 2018 | 1 | 0 | 1 | 0 | 0 | 0 | 2 |
| March, 2018 | 0 | 0 | 2 | 0 | 0 | 1 | 3 |
| April, 2018 | 1 | 0 | 2 | 0 | 0 | 0 | 3 |
| May, 2018 | 0 | 0 | 1 | 0 | 1 | 0 | 2 |
| June, 2018 | 1 | 0 | 1 | 1 | 0 | 0 | 3 |
| August, 2018 | 0 | 2 | 0 | 0 | 0 | 0 | 2 |
| January, 2019 | 0 | 0 | 2 | 0 | 1 | 0 | 3 |
| February, 2019 | 0 | 0 | 2 | 0 | 1 | 0 | 3 |
| March, 2019 | 0 | 0 | 0 | 1 | 0 | 0 | 1 |
| **Total** | 7 | 2 | 29 | 8 | 4 | 8 | **58** |

see and interpret the situation. We propose that using communication strategies and changing the role of agents in a story might affect the readers' perception of the reality and of the companies' role relating to the breach.

## 3 COMMUNICATION STUDY

We analyzed the Equifax [7] data breach because it happened recently (the Equifax's data breach occurred on May 2017, with investigation and analyses into 2019), and is of great importance due to the following reasons: the significant role of Equifax on people by impacting their access to many necessities, the extreme sensitivity of financial information breached and risk of identity theft, the centrality of data security to Equifax – its primary mission is stewardship, and the comprehensive analysis of the causes showing how they were at fault.

We used two categories of sources: "controlled news" issued by the company, and the other sources which contained media items, and government documents. The data presented in the media category includes technical news sites like *The Register* (a British tech and science news service) and the *HackRead* website (that centers on technology, security, privacy, surveillance, cyber warfare, cybercrime and first-hand hacking news), and reputable newspapers (i.e. *New York Times*, *The Washington Post*, *The Guardian*). The government documents category includes the United States Government Accountability Office (GAO) report to congressional requesters, the Congress of United States letter to the Federal Trade Commission and the Office of Management and Budget and the Equifax Minority Report describing the details of the cybersecurity attack. To retrieve all the stories concerning the data breach, we first searched the company's press releases archive. We then selected reputable news sites due to their national circulation and impact, and searched for keywords- "Equifax data breach". Some stories were excluded due to duplication or unrelated items such as the story of another data breach where it referred to the Equifax data breach. The analysis covers the period from September 2017 until March 2019 (see Table 1).

## 3.1 Coding

We first analyzed the sources using Image Repair Theory because it forms the basis of different approaches in the field, and it is best suited to a rhetorical method that examines texts produced by a company to identify their language and discourse patterns . In the context of a data breach, responsibility can appear in different forms, a company can be blamed for a poorly performed action that hurts consumers or neglect like poor security practices or flaws in a system that allows a breach. Perception of responsibility and offensiveness of an action can seem more important than reality, so businesses try

to use different strategies to affect the perception of responsibility. The coding sheet for image repair included the following strategies as nodes: **shift the blame**, **defeasibility**, **accident**, **bolstering**, **minimization**, **compensation**, **corrective action**, and **mortification**.

We used NVivo12, which is a qualitative analysis tool, to code our data. We first imported all the documents into the software, with different folders for different sources. We then created our first group of nodes for image repair strategies; we created a case node to represent each strategy, and we gathered references by coding sources at the nodes.

We then used the narrative-semiotic method of analysis [15] to understand how the positioning of agents changes when the narrator changes. We used this method to clarify how the company and the media communicate with the general public to direct their sense-making process. Based on the actantial model [8], the six role categories were applied to analyze the documents. To find the patterns in different sources we used the following categories as our nodes in NVivo12: the sender, the subject, the object, the receiver, the helper, and the opponent [15]. IRT and narrative-semiotic were used as a framework to explore the full text content of each document. One researcher did the coding and the results were reviewed by two other researchers.

## 3.2 Overview

To begin, we present a short description of major events and actions in the Equifax story. We divided Equifax's story into a series of narrative episodes which refer to different stages of the narrative trajectory. The information in this story was taken from press releases posted on the Equifax's official site [7] and the GAO report [17].

### 3.2.1 The Initial situation

Equifax is a consumer credit reporting agency. Equifax collects and aggregates information on over 800 million individual consumers and more than 88 million businesses worldwide, and its database includes employee data contributed from more than 7,100 employers [7].

### 3.2.2 The Attack

On 7 September 2017, Equifax announced a cybersecurity incident impacting approximately 143 million U.S. consumers and an unknown number in the UK and Canada. Hackers exploited a U.S. website application vulnerability to gain access to certain files in mid-May 2017. They stayed on the system until they were detected in July 2017.

The information accessed primarily includes names, Social Security numbers, birth dates, addresses and, in some instances, driver's license numbers. In addition, credit card numbers for approximately 209,000 U.S. consumers, and certain dispute documents with personal identifying information, for approximately 182,000 U.S. consumers, were accessed. As part of its investigation of this application vulnerability, Equifax also identified unauthorized access to limited personal information for certain UK and Canadian residents.

According to Equifax officials, beginning on May 13, 2017, attackers gained access to the online dispute portal (which maintained documents used to resolve consumer disputes) and used a number of techniques to disguise their activity. They extracted a portion of the PII (Personally Identifiable Information) residing on the systems.

After successfully accessing the information, the attackers exfiltrated the data in small increments, using standard encrypted web protocols to disguise the exchanges as normal network traffic. The attack lasted for about 76 days before it was discovered.

### 3.2.3 The Response

Equifax officials stated that, on July 29, 2017, approximately 2.5 months after the attackers began extracting sensitive information, security personnel conducting routine checks of the operating status

and configuration of IT systems detected the intrusion on the online dispute portal. A misconfiguration due to an expired digital certificate was the reason the intrusion was not noticed before. Equifax then blocked several Internet addresses from which the requests were being executed to try to stop the attack. The IT department discovered a vulnerability in the Apache Struts web application framework as the initial attack vector. The US-CERT had notified the company about this vulnerability before this incident. The Apache Foundation also had reported the vulnerability (CVE-2017-5638 [1]) in early March 2017.

Equifax took the website offline and then took steps to identify the stolen data and the number of affected people by this incident. Once Equifax officials found out how the attackers were able to access to the company's databases, they took measures to address this problem and avoid it in future. For the challenging task of identifying the affected individuals, Equifax compared the affected database with company's internal databases that were not impacted by the data breach.

On September 7, 2017, Equifax stated in its press release that the company had set up a dedicated website to help individuals determine if their information might have been exposed in the breach. Additionally, Equifax reported that it would provide several services to all U.S. consumers, regardless of whether their information had been compromised, free of charge for one year.

After the investigation, the company notified all U.S. state attorneys general regarding the approximate number of potentially affected residents in each state and its plans for consumer remediation. On March 1, 2018, Equifax stated that, overall, 2.4 million U.S. consumers whose names and partial driver's license information were exposed, had been identified.

### 3.2.4 The Resolution

The GAO report reveals how Equifax failed to protect Americans' personal data. According to the GAO, "Equifax determined that several major factors had facilitated the attackers' ability to successfully gain access to its network and extract information from databases containing PII," and that "key factors that led to the breach were in the areas of identification, detection, segmentation, and data governance."

Finally, the GAO's report highlights the critical need for legislation to protect consumers whose data is not adequately safeguarded, such as Senator Warren's and Senator Mark Warner's bill to hold credit reporting agencies like Equifax liable for data breaches. Under this legislation, Equifax would have paid at least $1.5 billion in penalties for the data breach.

## 4 ANALYSIS

### 4.1 Image Repair

We evaluated the company's press releases during the crisis using IRT. See Table 2 for the strategies used in press releases and illustrations of them. This table suggests that Equifax used the image repair strategies to minimize the apparent consequences of the data breach. Since the company was believed to be responsible for the incident, the CEO blamed the entire situation on IT staff who had not installed an Apache Struts patch issued in the weeks before the hack, and on technology failures. Moreover, the company used other strategies like: bolstering, compensation and corrective actions to reduce the offensiveness of the data breach. As a final general strategy, the CEO apologized to victims.

---

[1]"The Jakarta Multipart parser in Apache Struts 2 2.3.x before 2.3.32 and 2.5.x before 2.5.10.1 has incorrect exception handling and error-message generation during file-upload attempts, which allows remote attackers to execute arbitrary commands via a crafted Content-Type, Content-Disposition, or Content-Length HTTP header, as exploited in the wild in March 2017 with a Content-Type header containing a #cmd= string" [6].

Table 2: Image repair strategies in Equifax press release based on Testimony of CEO[2][3]

| Strategy | Key Characteristic | Illustration |
|---|---|---|
| Shift the blame | Another person did the act | The breach occurred because of both human error and technology failures. The human error was the individual who is responsible for communicating in the organization to apply the patch, did not do that. |
| Bolstering | Focus on positive feelings | Equifax was founded 118 years ago and now serves as one of the largest sources of consumer and commercial information in the world. |
| Compensation | Reimburse victim | A free credit file monitoring and identity theft protection package for all U.S. consumers. That includes free: 1) credit file monitoring by all three credit bureaus; 2) Equifax credit lock; 3) Equifax credit reports; 4) identity theft insurance; and 5) Social Security Number "dark web" scanning for one year. |
| Corrective action | Plan to solve or prevent problem | We set out to notify American consumers, protect against increased attacks, and remediate and protect against harm to consumers. In recent weeks, vulnerability scanning and patch management processes and procedures were enhanced. We took data security and privacy extremely seriously, and we devoted substantial resources to it. Equifax is doing everything in its power to prevent a breach like this from ever happening again. |
| Mortification | Apologize | I am here today to apologize to the American people myself and on behalf of the Board, the management team, and the company's employees. To each and every person affected by this breach, I am deeply sorry that this occurred. I sincerely apologize. I will close by saying again how so sorry I am that this data breach occurred. |

## 4.2 Narrative-semiotic (Equifax sources)

The second level of analysis involves the narrative-semiotic approach. Here we classified the agents in specific roles as they interact in the sequences of actions. Since a series of events can be told differently, the identification of the agents would change according to the role of the narrator.

### 4.2.1 The Initial Situation

The initial document that was considered for the narrative analysis was "Prepared testimony of Richard F. Smith before the U.S. House Committee on Energy and Commerce Subcommittee on Digital Commerce and Consumer Protection". The narrative story starts with the talk of the CEO of Equifax as a narrator about the initial situation of the company and how Equifax offered several services to its customers. Consider the following extract from the aforementioned text:

> FROM PREPARED TESTIMONY OF RICHARD F. SMITH:
>
> Equifax was founded 118 years ago and now serves as one of the largest sources of consumer and commercial information in the world. That information helps people make business and personal financial decisions in a more timely and accurate way. Behind the scenes, we help millions of Americans access credit, whether to buy a house or a car, pay for college, or start a small business. During my time at Equifax, working together with our employees, customers, and others, we saw the company grow from approximately 4,000 employees to almost 10,000. Some of my proudest accomplishments are the efforts we undertook to build credit models that allowed and continue to allow many unbanked Americans outside the financial mainstream to access credit in ways they previously could not have. Throughout my tenure as CEO of Equifax, we took data security and privacy extremely seriously, and we devoted substantial resources to it.

In this extract, customers (general people or businesses) are described as subjects who want to buy a house or a car, or start a small business. Equifax stepped into the role of helper to give information and to help people make business and personal financial decisions, and millions of Americans are described as receivers. Fig. 1(a) shows the actantial model inferred from this text.

### 4.2.2 The Attack

In the second stage of Equifax's story, a complication happened due to an external threat (attackers). Consider, for example, this part of the story where a narrator describes how the complication happened:

> FROM PREPARED TESTIMONY OF RICHARD F. SMITH:
>
> We now know that criminals executed a major cyberattack on Equifax, hacked into our data, and were able to access information for over 140 million American consumers. (...) Based on the investigation to date, it appears that the first date the attacker(s) accessed sensitive information may have been on May 13, 2017. The company was not aware of that access at the time. Between May 13 and July 30, there is evidence to suggest that the attacker(s) continued to access sensitive information, exploiting the same Apache Struts vulnerability. During that time, Equifax's security tools did not detect this illegal access.
>
> On July 29, however, Equifax's security department observed suspicious network traffic associated with the consumer dispute website (where consumers could investigate and contest issues with their credit reports). In response, the security department investigated and immediately blocked the suspicious traffic that was identified. The department continued to monitor network traffic and observed additional suspicious activity on July 30, 2017. In response, they took the web application completely offline that day. The criminal hack was over, but the hard work to figure out the nature, scope, and impact of it was just beginning.

The narrator of this press release proposes an attacker as the subject, and accessing information is the object. The Apaches Struts vulnerability and Equifax's security tools that couldn't detect the illegal access are both helpers that assist the attacker. The opponent category in this piece of story includes: the security department investigation, blocking the suspicious traffic, network monitoring, and taking the web application offline. The narrator (the company's CEO) emphasized the company's corrective actions to hinder the

---

[2]Prepared Testimony of Richard F. Smith before the U.S. House Committee on Energy and Commerce Subcommittee on Digital Commerce and Consumer Protection, 3 October 2017

[3]Oversight of the Equifax Data Breach: Answers for Consumers, Streamed live on Oct 3, 2017. https://www.youtube.com/watch?time_continue=16&v=4pgg2LCY8iE

attacker in the narrative quest. The following extract focuses on the helper role of CERT regarding notifying the companies on the vulnerability that could prevent the whole incident if acted promptly: "On March 8, 2017, the U.S. Department of Homeland Security, Computer Emergency Readiness Team ("U.S. CERT") sent Equifax and many others a notice of the need to patch a particular vulnerability in certain versions of software used by other businesses.". Equifax had 5 days to patch the vulnerability before the first date when attackers accessed sensitive information. Since the attacker could exploit the vulnerability of the Apaches Struts, the notification of CERT was a helper to the attacker too (See Fig. 1(b)).

### 4.2.3 The Response

In the second stage of Equifax's story, Equifax's actions to defend against the intrusion are described in the actions taken to address the complication. Therefore, according to the story, the agent's categories change in the narrative. The story starts with the repair efforts of the company, as the CEO as a narrator confessed that they failed to protect American consumer data and apologized for the act of data breach. The narrator (Richard F. Smith — Equifax's now retired CEO) continued to describe certain actions regarding how this incident happened.

> FROM PREPARED TESTIMONY OF RICHARD F. SMITH:
>
> Americans want to know how this happened and I am hopeful my testimony will help in that regard. As I will explain in greater detail below, the investigation continues, but it appears that the breach occurred because of both human error and technology failures. These mistakes – made in the same chain of security systems designed with redundancies – allowed criminals to access over 140 million Americans' data.
> Upon learning of suspicious activity, I and many others at Equifax worked with outside experts to understand what had occurred and do everything possible to make this right. Ultimately we realized we had been the victim of a massive theft, and we set out to notify American consumers, protect against increased attacks, and remediate and protect against harm to consumers. We developed a robust package of remedial protections for each and every American consumer – not just those affected by the breach – to protect their credit information. The relief package includes: (1) monitoring of consumer credit files across all three bureaus, (2) access to Equifax credit files, (3) the ability to lock the Equifax credit file, (4) an insurance policy to cover out-of-pocket costs associated with identity theft; and (5) dark web scans for consumers' social security numbers.

In this extract, Equifax is foregrounded as the main agent, occupying four positions. Equifax is described as a sender dictating to its employees and outside experts to make the suspicious activity right. Equifax also stepped into the role of receiver, as well as helper. Opponents in this part are criminals, human errors and technology failure. Fig. 1(c) shows the actantial model inferred from this text.

Overall, Fig. 1 shows the actantial model of the Equifax's story and we can see how the substories are connected to each other. The customer is the subject who wants to buy something or start a business as object, and Equifax is its helper to achieve the goal. In the other part of the story Equifax stepped into the role of subject who wants to solve the data breach incident, Equifax protection activity is a helper here, and human error, computer failure and attackers are playing the role of opponent. In the last part of the story, attackers are subjects who want to find access to the personal information of Equifax's customers, human error and computer failure is a helper and Equifax protection activity is an opponent.

### 4.3 Narrative-semiotic (other sources)

The other documents were the news provided in the technical websites, general news, and the GAO report. After coding these docu-

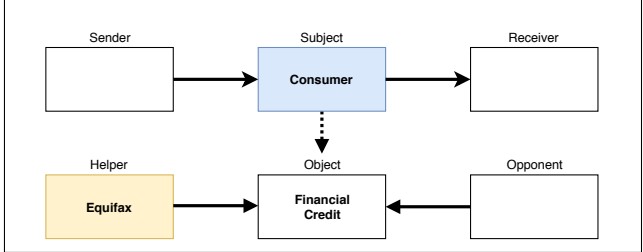

(a) The Initial situation, Equifax press Release

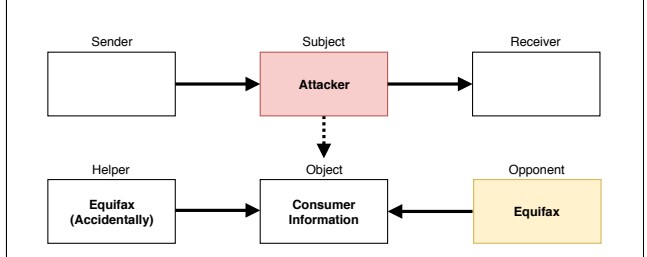

(b) The Attack, Equifax press Release

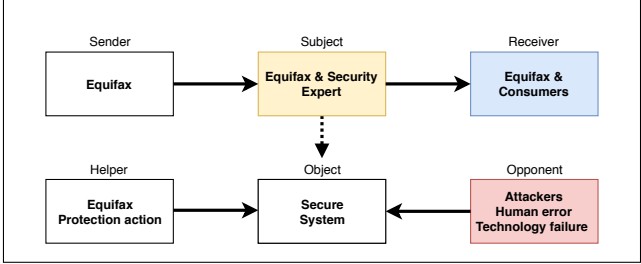

(c) The Response, Equifax press Release

Figure 1: Actantial model from Equifax press releases

ments using narrative components, we found two narrative programs or mini-narratives that were common among these texts; one from the attack stage of the story and the other from the response part of the story.

### 4.3.1 The Attack

In the first narrative episode, the attacker is the subject, and getting access to the information is the object. The company's failure to patch the flaw, an expired certificate, lack of notice of malicious activity, and lack of restriction on the database are all helpers to the attacker. The opponent category includes locking down the system so that the attackers would not be able to misuse the vulnerability and hiring an expert security team. Fig. 2(a) shows the actantial model of this mini-narrative.

### 4.3.2 The Response

In the second narrative, Equifax is the subject who wants to solve the incident (see Fig. 2(b)). The helper category in this mini-narrative that was extracted from the news includes the following components: log files, and new regulations.

Numerous data security failures such as the insecurity of Equifax's web setup, failing to patch the flaw promptly, the lack of restrictions on the frequency of database queries are the first element in the category of opponents. The second one is failure in notifying the data breach victims, the evidence that was extracted from different texts are as following:

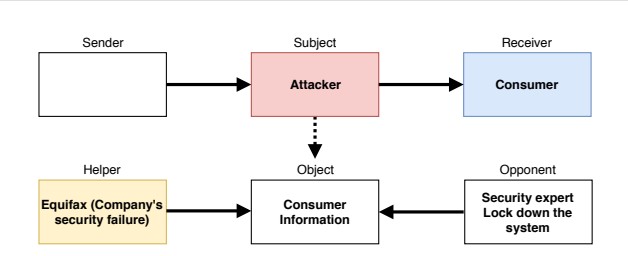

(a) The Attack, News and GAO report

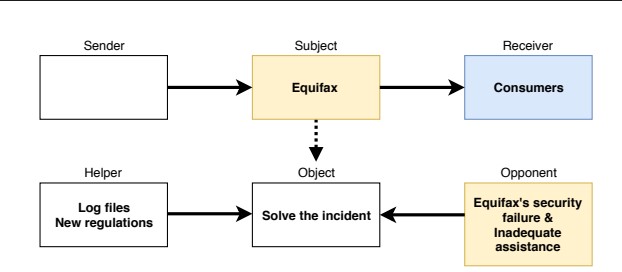

(b) The Response, News and GAO report

Figure 2: Actantial model from News and GAO report

FROM GAO EQUIFAX REPORT:

- Equifax executives - including its Chief Security Officer and Chief Executive Officer - kept the public in the dark for more than a month after they found out about the security intrusion.

- The attack lasted for about 76 days before it was discovered.

- Equifax and other big credit reporting agencies keep profiting off a business model that rewards their failure to protect personal information.

- In the three years before the Equifax data breach, the company spent only about 3% of its operating revenue on cybersecurity—less than the company spent on stock dividends.

FROM THE GUARDIAN:

- because of a process failure in 2016 that meant a limited amount of UK data was stored on the US system between 2011 and 2016.

- Congressman *Frank Pallone* said Equifax had an "ongoing lax attitude when it comes to protecting consumer data.

- An inadequate response to a data breach that included the personal information of up to 143 million Americans.

FROM THE REGISTER:

- Equifax was breached in "mid-May" 2017, realized it in July and got around to telling the world in early September.

- Apache Struts was popped, but company had at least TWO MONTHS to fix it.

- specifically, the lack of restrictions on the frequency of database queries allowed the attackers to execute approximately 9,000

such queries – many more than would be needed for normal operations.

- "As your company continues to issue incomplete, confusing and contradictory statements and hide information from Congress and the public, it is clear that five months after the breach was publicly announced, Equifax has yet to answer this simple question in full: what was the precise extent of the breach?" *Senator Elizabeth Warren* fumed in a missive late last week.

The third element in the opponent category is inadequate assistance in resolving the problem, such as no definitive action to hold Equifax accountable, betraying stakeholders by top Equifax executives, an overwhelmed call center and customer response site. Some examples of evidence found in the documents are:

FROM THE GUARDIAN:

- It is always a company's responsibility to identify UK victims and take steps to reduce any harm to consumers.

- Ying used confidential information to conclude that his company had suffered a massive data breach, and he dumped his stock before the news went public.

FROM THE REGISTER:

That was a hastily constructed WordPress bodge job, and victims were initially asked to agree to take any dispute to arbitration and forfeit the right to take part in any class-action lawsuit.

FROM GAO EQUIFAX REPORT:

- Equifax also failed to provide consumers full protection from new account identity theft.

- These consumer complaints included improper use of credit reports, incorrect information on credit reports, inadequate assistance in resolving problems, and problems with Equifax credit monitoring, fraud alerts, and security freezes in the wake of the breach.

Our analysis shows that the narrative programs extracted from company's press release are different from the ones extracted from news. More details will be discussed later.

## 5  QUESTIONNAIRE STUDY

### 5.1  Methodology

As explained above, our text analysis showed important differences between how organizations described data breach incidents, and how they were described by others. In particular, the texts from organizations appeared to position themselves in ways that might affect how readers perceived their role relating to the breach. We speculated that this might influence users' understanding of the data breach, and of organizational accountability. To explore this, we conducted a questionnaire study, where we asked participants to first read data breach descriptions and then asked their perception of various aspects of the breach.

We recruited participants through TurkPrime, which is an online crowdsourcing research platform that integrates with Amazon Mechanical Turk (MTurk) [12]. Participants were asked some questions about demographics and data breach basics, then read two short extracts describing data breaches, and finally responded to follow-up statements about the incidents described. The study took approximately 15 minutes, and we reimbursed participants $2 for their time. The study was reviewed and cleared by our Research Ethics

Table 3: Participants' demographics

| Age | % | Education | % |
|---|---|---|---|
| 18-30 | 36% | Less than High school | 0% |
| 31-40 | 42% | High school degree or equivalent | 30% |
| 41-50 | 13% | College/ Bachelor's degree | 54% |
| 51-60 | 3% | Trade or technical degree | 5% |
| Over 60 | 6% | Graduate degree | 11% |

| Field of study | % |
|---|---|
| Formal science (computer science, Logic, Math) | 15% |
| Natural science (Biology, Physics, Chemistry. . . ) | 10% |
| Social science | 16% |
| Engineering | 9% |
| Arts | 10% |
| Law | 3% |
| Other | 37% |

Table 4: Wilcoxon tests on responses to each statement (company vs. news).

| Statement | P-value |
|---|---|
| 1. The customers of the company are victims of the data breach. | n.s. |
| 2. The company is the victim of the data breach. | $p < 0.01$ |
| 3. The company had a relaxed attitude about protecting customers data. | $p < 0.01$ |
| 4. The company takes security measures seriously. | $p < 0.01$ |
| 5. Attackers wanted to harm the company. | n.s. |
| 6. Attackers wanted to harm the customers. | n.s. |
| 7. The company is helping customers recover from the breach. | $p < 0.01$ |
| 8. The company put customers at risk by neglecting data protection. | $p < 0.01$ |
| 9. The company is accountable for problems resulting from the data breach. | $p < 0.05$ |
| 10. The customers are accountable for problems resulting from the data breach. | n.s. |

Board. We recruited 100 participants, specifying participants must be residents of the US or Canada. By far most were from the US, 96, with only 4 from Canada, with 33 female and 67 male. A summary of their demographics is shown in Table 3.

There were two pairs of extracts about the Equifax data breach. Each pair contained one extract from the company itself, and one from another source (e.g., news), but no company names were mentioned. We extracted each text from different documents where the company used the communication strategies, both extracts had the same topic and length. Participants were shown a pair at random, and the order within the pair was also random. We randomized the pairs, and the order as a practical approach to balance, and later confirmed they were indeed reasonably balanced. After each extract, participants were shown 10 statements, and responded using a Likert scale to gauge their perception about the breach motivations, company security measures, after-breach issues, and responsibility. Each of the responses were scored 1-5, where 1 stands for "Strongly disagree", 2 for "Somewhat disagree", 3 for "Neutral", 4 for "Somewhat agree", and 5 for "Strongly agree".

Our hypothesis were that responses to each of the 10 statements would differ by the source of the text read by the participants. We examined responses, removing those unrealistically quick, or with inconsistent answers between similar questions. Then, we analyzed the results of the questionnaire by calculating the median and spread of the distribution of our data. We also did Wilcoxon tests to see if the source (company or news), order of text within each group, and the pair of data breach description affected the participants' responses.

## 5.2 Results

Fig. 3 shows the ten statements together with boxplots in describing the responses to each statement. For each statement, there are two boxplots, one for responses to the company text, and the other for responses to the news media text.

### 5.2.1 Victimhood

After reading both the company text and the news text, in response to being asked who the victim in an Internet data breach is, participants strongly agreed that customers of the company are the victims (Mdn= 5/5; later numbers also refer to medians). However, the result for the two texts was quite different when asked if the company is the victim of the data breach. After reading the company's description of the breach, participants agreed (4/5) that the company is also a victim, but reading the news shows a slightly different result, but they again agreed that the company could also be a victim (4/5).

### 5.2.2 Company's attitude

When asked about the company's attitude regarding data protection, a considerable difference can be seen between the two texts. Par-

ticipants reading the company's description disagreed (2/5) that the company had a relaxed attitude about protecting customers data, and they agreed that the company took security measures seriously (4/5). However, participants reading the news text showed different results, and agreed (4/5) that the company had a relaxed attitude and they disagreed that the company took security measures seriously (2/5).

### 5.2.3 Intent to harm

In response to being asked what the attackers' purpose was, we got quite similar results. They agreed that attackers wanted to harm customers as well as the company (4/5).

### 5.2.4 Helper or opponent

Participants reading the company's text agreed (4/5) that the company was helping customers to recover from the breach. They somewhat agreed (4/5) that the company put customers at risk by neglecting data protection. Those reading the news text, however, strongly agreed that the company put the customer at risk (5/5), and they disagreed that the company was helpful in after-breach actions (2/5).

### 5.2.5 Accountability

In response to being asked about the accountability, the results of reading both the company and news' text were quite similar but significantly different at 0.05 level. Most participants believed that the company was accountable for problems resulting from the data breach (Company= 4/5, News= 5/5), and the customers were not responsible (Company= 1/5, News= 2/5).

### 5.2.6 Hypotheses

Our hypotheses were that there would be a different response for those reading the company's text and those reading the news text about a data breach. To test our hypotheses, we did Wilcoxon tests for the responses to each statement, choosing this non-parametric test because the data was ordinal. The results are shown in Table 4. We can see that the company's and the news description make a significant difference in participants' responses about victimization, the company's security measures, its attitude in data protection, and helpfulness in after-breach actions (these are marked with red boxes in Fig. 3).

We had two pairs of data breach descriptions (company and news), one of the pairs was assigned randomly to each participant, and the order within the pair was also randomized. For each statement we used Wilcoxon tests to see if the pair (one of two) and the order (company first and news first) affected the participants responses.

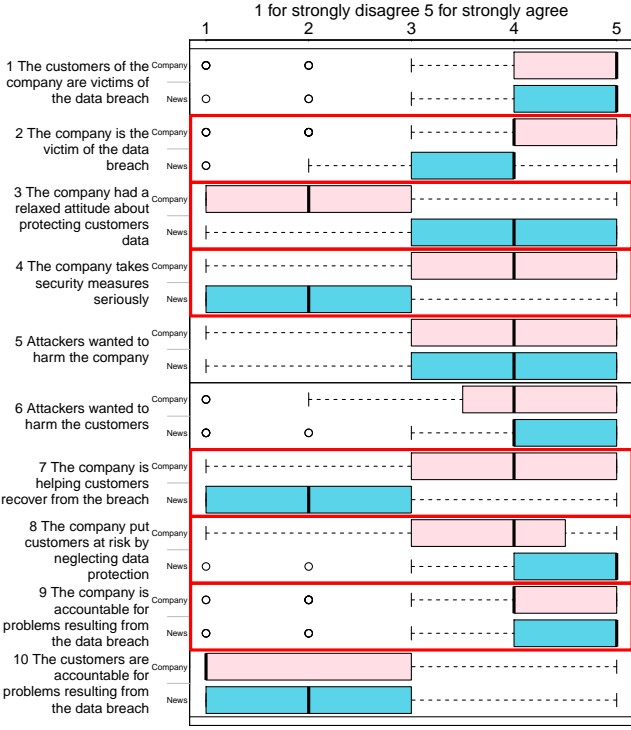

Figure 3: Data breach description (pink= Company, blue= News, red boxes= where the results are significantly different between Company and News), vertical lines through the boxes are median

The results showed that the order of texts did not change the participants responses. Which pair of texts also made no difference, except for the statement about whether the company is a victim or not and the helpfulness of about-breach actions. Although we used non-parametric tests, we also checked a multi-way ANOVA test for the pairs, order, and source of texts, and the results confirmed our findings with Wilcoxon tests with little difference.

### 5.2.7 Interpretation

In this study, we showed half our participants text about a data breach from a company, and half text about the same breach from a news source. We then asked them to respond to 10 statements about the breach, and we expected that their responses would differ for all 10 statements depending on the source of the text.

For most statements (2, 3, 4, 7, 8, 9) this is what occurred. For others, however, we saw little difference (1, 5, 6, 10). The overall patterns of participants' responses are shown in Fig. 3. For statement 1 (victimhood of customers), it seems that both sources led participants to feel that the customers were indeed victims, whereas statement 2 shows that people who read the company text more strongly felt that the company was also a victim. However, statements 5 and 6 (who the attackers wanted to harm) presented no difference in responses, despite their similarity to statements 1 and 2. We speculate that participants perceived a difference between the concept of victimhood and the concept of intent to harm. Both customers and company might, or might not, be victims, but *intent* to harm was more difficult to distinguish.

For statement 9 (accountability of company), those who read the company text were less strong in feeling the company was accountable for the breach, but for statement 10 (accountability of

customers), there was agreement that customers themselves were *not* accountable. Both these seem understandable. For statement 9, both sources suggest that the company has some level of accountability, but the non-company text makes it seem more clear.

For statement 10, neither source suggests that customers have any accountability. On reflection, the issue of accountability has a subtlety that needs to be addressed. It is clear the attackers have a key role in the breach, so they can be seen as having accountability. The company, however, has a duty of care, and any failing in that duty might also be seen as involving accountability. In the physical world, burglars might rob a bank, but if it emerged that the bank left their doors unlocked at night, we suggest any bank customer would regard them as also accountable.

## 6 DISCUSSION

The primary goal of our work was to explore how organization communications about data breaches might affect user perception. To do this, we first studied the nature of the communication itself. Using Image Repair Theory, we analyzed press releases posted on official company websites. We found that Equifax press releases had characteristics consistent with tactics to reduce reputational damage and therefore financial loss. Recognizing that the way the news media frames a crisis might be different to the framing in an organization's press releases, we next explored that issue. We used techniques from narrative-semiotic to examine the structure of the stories being told, and found that the agents were not positioned the same way.

Considering the first narrative story studied, our comparison of the Equifax press releases with news and GAO reports shows important differences with respect to the positioning of Equifax (see Fig. 4). In the press releases, there was emphasis on Equifax as a helper, presenting the company's protection actions. In the news and GAO reports there was emphasis on Equifax as an opponent, presenting the company's weak security protection of consumer data. Moreover, the news media had a focus on the company and its security failure, whereas the company appears to use scapegoating as its primary crisis response strategy [4] and suggesting responsibility lay with a single unnamed IT staff member. The ethics of scapegoating is doubtful [10], suggesting a manipulative approach used deflect responsibility. The company's apparently lax attitude in crisis response was heavily criticized by the media. The news text suggests Equifax shares responsibility for this incident. However, Equifax positioned itself as a receiver to emphasize it is a victim, a strategy that is consistent with an attempt to reduce its responsibility [5].

Equifax appears to map all their actions to the helper category, in a manner consistent with Image Repair Theory. For example, a bolstering strategy places the company in a helper position, deflecting responsibility by shifting the blame, scapegoating puts the company in victim position, and compensation strategies stress the company acts as a helper. However, when the news media narrates the story, the mapping of the actions and agents goes to the opponent category, since the media is not concerned with Image Repair.

Our second step was to explore how the strategies used in the company press releases might influence the public understanding of data breaches. We conducted a questionnaire study to see if data breach incident descriptions from different sources, the company's and the news media, result in different perceptions of the incident. After reading the text extracted from the company's press release, participants tended to rate the company's after-breach action and security measures higher. They also thought that the company was helping their customers and did not put them at any risk. However, we got different results from participants who read the news texts reporting the same incident; participants disagreed that the company took the security seriously and their after breach protective actions were not acceptable to help the customers. The company was regarded as a victim after reading the company's description; however,

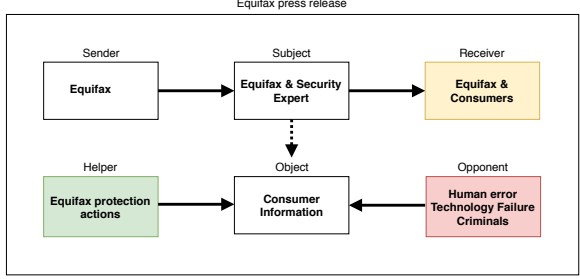

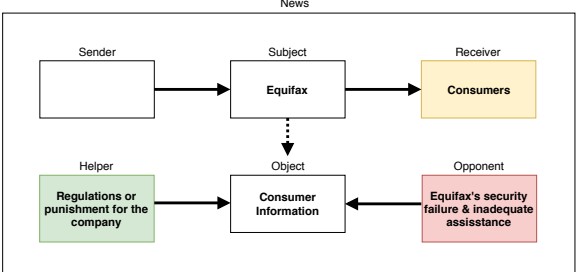

Figure 4: Equifax: comparing narrative program, The Response stage

the news approach in narrating the data breach resulted in a different perception of participants. This therefore confirmed our speculations based on our text analysis. It also confirms the effectiveness of IRT and its relevance in crisis communication.

Of course, it is not surprising that companies tend to present themselves in a better light that the news media. Nor is it surprising that they used strategies that have been developed to help them to do this. However, our text study shows that their Image Repair strategies exhibit some important characteristics. In particular, they show differences in how agency is presented, which, in turn affects readers' understanding of what happened.

### 6.1 Implications

Studies of HCI and security highlight the role of user mental models in understanding issues relating to security (e.g. [3, 20]), and we know many users continue to use breached services (see [1, 16]). Our study shows that organizations, by their communications strategies, may be contributing to users weak mental models.

One design implication might be improving software design to support users ability to track access to their data, with enough detail to help users determine provenance and legitimacy, perhaps with alarms for especially sensitive data. This would enable user engagement and oversight of their own information. This would also allow users to inform organizations of discrepancies, and knowledge of this new transparency would promote increased diligence by organizations. Moreover, organizations could make protective measures more explicit to users, both to assure users, and to emphasize their diligence. The potential effect of all such measures could be studied in future research. We should also acknowledge that some of the measures we suggest may require changes in underlying software architecture, but research could show that such changes are justified, and potentially imperative.

The possibility of crisis communications influencing user perception of accountability should be considered by communications professionals and legal scholars, to better establish the line between promoting the organization and misleading users.

### 6.2 Limitations

There are some limitations to address in this research. First, the communication study was focused on one case study, and the same

analysis on several data breaches may reveal different results. Second, the wording of each statement in the questionnaire study might cause the difference that participants perceived in the concept of victimhood and the concept of intent to harm.

## 7 CONCLUSION

In this paper, we presented our study on communication about data breach events which exposed private consumer data. We first analysed Equifax press releases and notifications to identify their strategies, and then analysed news stories and government reports on the same events; we studied 58 stories in all.

We found that the company used crisis communication strategies to reduce its reputational damage and financial loss. Our analysis also showed that there are differences between press releases, major newspaper and technical news when reporting the same data breach incident. In our narrative-semiotic analysis, we found the company mapped their after-breach actions into helper category; but the narrator of news reports mapped them into the opponent category. These narrative changes affected reader perception about these data breaches.

Our questionnaire study revealed that the dissimilar approach detected in document analysis when narrating the same story from a different point of view (companies and news) has a considerable influence on the general public's perception of a data breach incident.

Large scale data breaches are a serious matter, not just for organizations, but for the thousands or millions of users who have private data exposed, making them vulnerable to a range of consequences. Despite this, it is unclear if users understand what exactly has happened, where accountability lies, and how to proceed. In work on human factors in computer security, it has often been found that users have only weak mental models of online threats and defences, e.g. [3, 20]. When user data is exposed by a large scale data breach, communication with the user may well be primarily from the organization itself. Our research suggests that communication from organizations may misrepresent the data breach events leading to misleading perceptions of the crisis and the company's accountability. Design of software that stores sensitive personal information should support users in maintaining better awareness of data breaches.

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
