# OpenReview forum: "We're Here to Help: Company Image Repair and User Perception of Data Breaches"
_graphicsinterface.org/Graphics_Interface/2020/Conference — GI 2020_

### Official Review · AnonReviewer1 · 2020-01-09
**See review**

**Confidence:** 4
**Rating:** 6

**Review:**

This paper presents a study of user perceptions to communications about data breaches, based on a use case from a real example. The method applies Image Repair Theory to understand the strategies applied in the communications and narrative semiotics to model the examples of individual strategies. A user study conducted on mechanical turk shows a difference in perceptions for different communications sources - company or external sources.

This paper is very well written and quite interesting to read. The method and study are described in detail and the analysis and interpretation of the results are clear. The figures do a good job of demonstrating each of the examples discussed. Overall the study is sound and the results provide a contribution in an area that has apparently received little direct attention.

However, I do have a few concerns worth noting:
- my main concern is that the paper seems to have little relevance to HCI. The study primarily concerns communications and user perceptions, and the link to HCI is not evident. This comes a little more clear in the final pages of the paper. It would be better to make the connection very evident early in the motivation, however it is only mentioned very tangentially. Despite the later discussion, I am still not fully convinced that the paper would not be a better fit for a business or marketing conference, thus my rating does not reflect the overall high quality of the work.

- The study contains several random elements and could be better controlled. In general it is better to control variables rather than randomise them, for instance to ensure the intended balancing is achieved. I understand that this was probably done for practical reasons but these should be discussed along with the potential effects on the results.

- The paper does not mention whether participants were paid for the study. I note that the ethics review was passed, however, given that turk recruits paid workers, not volunteers, it is important to ensure that the participants are not exploited and paid a fair wage.

- the boxplots in figure 3 are difficult to interpret. I assume these show the median not the mean, but it would be good to clarify this. The whiskers add visual clutter rather than provide much additional information, so a more standard visualisation method would be preferred.

---

### Official Review · AnonReviewer3 · 2020-01-09
**Interesting paper, carefully written, HCI takeaways unclear**

**Confidence:** 4
**Rating:** 7

**Review:**

This paper presents a case study in crisis communications, examining the differences in the way news outlets and a company (in this case, Equifax) report about a large scale data breach, and how these different styles may affect the interpretations of the public.

The authors use several different techniques to analyze the data, including Image Repair Theory and Narrative Semiotics. These were not theories I was familiar with, and I found the paper introduced them well and the analysis was thorough and well-written. The theories present the press releases of Equifax in a very different light from the news reporting from a variety of sources. The coding was carried out by one researcher, but confirmed by two others, so while we don't have a measure of interannotator agreement, I think the method is suitable. The paper includes a lengthy narrative of the Equifax data breach, including the events leading up to it, the impact, and the response. The two theories are then applied to the statements of the company and the statements of others as listed in the news. The IRT framing helped me to understand better the strategies used by the company (table 2) in such a situation, and I think in future I will read and understand such statements with a more critical eye.

The paper also includes a survey of 100 Turkers who were asked to read statements from the company and the news (in random order) and complete a 10 item questionnaire. The researchers were looking at whether the framing by either the company or the news (or the order) would affect things such as if the reader feels the customer or the company was the victim. The results were interesting, in particular, that the framing may affect the way people interpret whether the company acted responsibly. I have some concerns about the study design, which should be addressed in a revision. In particular, how were the specific passages selected for the survey? There were two pairs of passages - were they matched somehow to be similar in length and topic? Why not use many more different passages?

The biggest negative for this paper from the point of view of the GI conference is the topic match and relevance. This is only tangentially an HCI paper. For example, none of the references are from HCI literature. The paper presents some ideas for HCI takeaways such as the need for tools to assist people in tracking the use of their data. To me this is weakly related to this work. The stronger takeaways as presented are for communications and legal professionals around crafting communications so that they are truthful and not manipulative. One HCI implication that I see is that perhaps we could automate some of the analysis here to provide tools for consumers to "see through" the manipulation of corporate (or news) messaging to support critical readings.

Overall, I think this is a solid paper but borderline out of scope for the conference.

Minor: some issues with spacing around punctuation - bottom left of p3

---

### Official Review · AnonReviewer2 · 2020-01-10
**Wrong conference to submit this paper to. But a nice paper.**

**Confidence:** 3
**Rating:** 6

**Review:**

In this paper, study how organization communications about breaches affect customer perceptions. The authors presented a study of the nature of the communication using image repair theory by analyzing press releases on company websites. It was revealed that the press releases were consistent with approaches to reduce reputational damage. Narrative semiotics were used to examine the stories’ structures in the media. It was found that the company was not depicted the same way. The authors next explored how the strategies used in the company press releases influence the public understanding of data breaches. This was done through a questionnaire which collected responses to see if data breach incident descriptions from different sources, the company’s and the news media, result in different perceptions of the incident. Participants were recruited through TurkPrime. The statistical test chosen (Wilcoxon) is the correct inferential statistical method for this study to the best of my knowledge. Findings are as expected: companies tend to present themselves in a better light that the news media.

The quality of presentation in this paper is very high. The main issue with the paper is that there is virtually no relevance to HCI. I am very surprised why authors chose this conference to submit a piece of work like this. The subject matter that is being explored fits much better with a business conference. It is an interesting paper to read though. I did not rate this paper high because of its low relevance to the theme of this conference. It does not reflect the quality of work.

---

### Meta-Review · Area_Chair1 · 2020-01-10

**Recommendation:** Accept
**Confidence:** 2

**Metareview:**

All the reviewers have agreed that the relevance of this work to Graphics Interface is very hard to justify.
R3 has some concerns about the study design which should be addressed.
R1 asks to clarify if participants got paid and gives some further minor suggestions to improve the paper.

---

### Decision · Program_Chairs · 2020-01-11

Accept